# Insecticidal and Repellent Activity of Plant Powders on the Weevil (*Sitophilus zeamais*) in Stored Corn Grains in a Rural Community of Oaxaca, Mexico

**DOI:** 10.3390/insects16030329

**Published:** 2025-03-20

**Authors:** Citlaly Peña-Flores, Arturo Zapién-Martínez, Gabriel Sánchez-Cruz, Leobardo Reyes-Velasco, Aristeo Segura-Salvador, Jaime Vargas-Arzola, Luis Alberto Hernández-Osorio, Honorio Torres-Aguilar, Héctor Ulises Bernardino-Hernández

**Affiliations:** Chemical Sciences Faculty, Benito Juárez Autonomous University of Oaxaca (UABJO), Av. Universidad S/N. Cinco Señores, Oaxaca de Juárez, Oaxaca 68120, Mexico; cpeaflores@outlook.com (C.P.-F.); zaarma@yahoo.com (A.Z.-M.); gsanchez@uabjo.mx (G.S.-C.); lreyes.cat@uabjo.mx (L.R.-V.); ass@aristeosegura.com.mx (A.S.-S.); vajcquabjo@hotmail.com (J.V.-A.); luisheol@hotmail.com (L.A.H.-O.); qbhonorio@hotmail.com (H.T.-A.)

**Keywords:** weevil, corn, bioinsecticide, repellency, plant powders

## Abstract

The weevil (*Sitophilus zeamais*) is one of the leading pest insects that attack corn grains in storage, causing considerable economic losses and their unavailability for consumption. Its control is carried out with the use of hazardous chemical pesticides. Given this situation, the use of plant species is attracting attention since they are ecologically safer than chemical solutions. Four plant powders were tested (epazote-*Chenopodium ambrosioides* L.-, oregano-*Origanum vulgare*-, hierba santa-*Piper auritum*- and laurel -*Laurus nobilis*-) for controlling of the weevil in corn stored in the rural community of Santa María Zacatepec, Oaxaca, Mexico. The epazote and hierba santa were the species that caused the most significant mortality and repellency. Both plants are a low-cost and accessible alternative since they are common and abundant species in the different regions of Oaxaca.

## 1. Introduction

Corn (*Zea mays*) is one of the most important crops worldwide due to its use as human and animal food (mainly in sub-Saharan Africa and Latin America), as well as for obtaining various industrial products [1]. It is the most representative crop for Mexico, given its economic and biosociocultural importance [2]. Like other cereals, corn has one of the most critical stages: storage. They are highly susceptible to attacks by various pests, mainly insects, which are responsible for damage to the grain, translating into considerable economic losses and affecting its availability for consumption [3]. Therefore, it represents one of the main problems small farmers face in rural environments [4]. One of the most harmful insects is the corn weevil (*Sitophilus zeamais*), which causes losses mainly in tropical regions, given the environmental conditions that favor its growth [5]. The damage originates when adults pierce the grain to lay eggs, while the emerging larvae generate grooves in the endosperm to feed [6]. It has been estimated that at the time of harvest in developing countries, a small percentage of grains may be contaminated by this insect; if not prevented or controlled, losses in the warehouse may range from 20% to 60% or higher in more serious cases [7]. Although the data are not precise for Mexico, post-harvest losses have been estimated to be between 10 and 40% of total production yearly. This particularly worries the rural population since their food security depends on their production [8].

To minimize losses, farmers use synthetic chemicals, including various contact insecticides (organophosphates or pyrethroids) and fumigants (methyl bromide or aluminum phosphide) [9]. Unfortunately, using these pesticides represents a risk to public health due to accidental oral overexposure to toxic residues in the grain or poisoning by inhalation since it is very common for the grain to be stored in homes, in addition to the pollution generated to the environment [10]. Although pesticides are reasonably effective, sooner or later, resistance to them appears, and using an increasing amount of pesticide not only increases costs but also accelerates the accumulation of the active ingredient in water, soil, air, and food, resulting in a consequent increase in risks to human and environmental health [11]. Therefore, searching for more economical, less dangerous, more readily available, and friendly options for health and the environment is necessary. An alternative is using natural products derived from plants, which are generally biodegradable and have a low environmental impact. The use of plant species with bioinsecticidal properties has gradually spread in recent decades in Latin America, with promising results in the control of various agricultural pests [12]. Most plant species with insecticidal and fumigant properties are very aromatic due to secondary metabolites mainly from the terpene group, which have shown toxic properties on some insect pests [13].

Products of plant origin used as bioinsecticides have been essential oils, extracts, or powders. Powder presentation is the simplest way to protect stored grains, obtained by drying and grinding the plant to mix it with the grain later. The plant species used in powders are characterized by the presence of specific metabolites grouped into phenolic compounds, terpenoids, and alkaloids [14], which have insecticidal, larvicidal, repellent, growth inhibition, and oviposition deterrent properties [15]. Among the plants whose bioinsecticidal effects haves been documented are epazote (*Chenopodium ambrosioides*) [16], oregano (*Origanum vulgare*) [17], hierba santa (*Piper auritum*) [18], and laurel (*Laurus nobilis*) [19]. These species are very popular in Mexican cuisine and are grown throughout the national territory. Among its medicinal uses, *C. ambrosioides* is used mainly for gastrointestinal dysfunctions and worm parasitosis [20]. *O. vulgare* is used as a diuretic, gastrointestinal analgesic/anti-inflammatory, and in some respiratory conditions [21]. *P. auritum* is used for stomach pain, diarrhea, and colic [22]. *L. nobilis* is used as an epigastric anti-inflammatory, antiflatulent, and to relieve hemorrhoids and rheumatic pain [23]. In the Sierra Sur of the State of Oaxaca, Mexico, specifically in the municipality of Santa María Zacatepec, the weevil is a common problem in the storage of corn, and the plants above are abundant in the region. The objective of this study was to evaluate the insecticidal and repellent effects of four plant powders (*C. ambrosioides*, *O. vulgare*, *P. auritum*, and *L. nobilis*) in the control of adults of *S. zeamais* in stored corn in the aforementioned rural town.

## 2. Materials and Methods

### 2.1. Place Where the Study Was Carried Out

The study was carried out in the community of Santa María Zacatepec, located in the Sierra Sur region of Oaxaca, Mexico (Figure 1) (Latitude: 16°45′ N and Longitude: 97°59′ W, 340 m above sea level, climate warm, average annual temperature of 32 °C, average yearly rainfall of 2000–2500 mm) [24]. The population was 17,100 until 2020 [25], of which 85.7% were in poverty [26]. The plant material and the insects were prepared in a local home with the owner’s consent. The mortality and repellency experiments were carried out in a room of the same house where the corn was stored, following the methodology proposed by Lagunes and Rodríguez used in various studies [27,28,29,30].

### 2.2. Obtaining and Preparing Plant Material

The plant species were collected in the backyards of various homes in the community of Santa María Zacatepec, Oaxaca, with prior consent from the owners of those homes. For each species, 0.5 kg of fresh leaves were weighed using an electronic scale (TECNOCOR PPR-40, Tecnocor, Puebla, Mexico) and dried on paper in a dry, clean place away from direct sunlight, at room temperature (between 26 °C and 32 °C, measured using a glass thermometer) for 7 days. After drying, they were crushed in a hand mill, subsequently ground in a ceramic mortar until a fine powder was obtained and sieved with a 297 μm stainless steel laboratory mesh. The powders obtained were stored in plastic containers at room temperature for preservation before the experiments. The plant species were deposited in the Herbarium of the Faculty of Sciences of the National Autonomous University of Mexico, with the following registration numbers: 186310 (*O. vulgare*), 186312 (*P. auritum*), 186313 (*L. nobilis*), and 186290 (*C. ambrosioides*).

### 2.3. Obtaining and Preparing Corn

The substrate used for feeding the insects, both in their reproduction and bioassays, was native white corn. A total of 25 kg were collected in storage conditions in the community of Santa María Zacatepec, which was exposed to the sun for 8 h and sifted with a sieve to eliminate residues. The temperature of the concrete was 46.6 ± 1.1 °C, measured with a digital infrared thermometer between 2 and 3 p.m. This procedure guaranteed that the corn was free of eggs and adult specimens of the weevil. This procedure is a traditional method to prevent insect attacks since they do not tolerate temperatures higher than 40 °C, mainly if a concrete floor is used [31].

### 2.4. Obtaining and Preparing the Insect

The insects were collected in homes in the community of Santa María Zacatepec, where the stored corn had no product applied for its control. Corn samples were sieved (0.5 cm opening) to separate adult weevils and placed in a plastic container covered with organza fabric to allow them to breathe. To reproduce the insect, 50 individuals were placed in a transparent plastic jar with a capacity of 3 L. As for food, 2 kg of previously treated white native corn was placed. The jar was covered with organza fabric to allow ventilation inside and prevent insects from escaping. For this activity, two jars were used, which were placed in a cool, dry space inside the home at room temperature (between 26 °C and 32 °C) for 30 days (approximate time of the weevil’s life cycle) to favor the reproduction, oviposition, and maturation of the required adults. Mortality and repellency tests were carried out when the adults (aged between 7 and 14 days) were obtained. Previously, the specimens were identified as *S. zeamais* with the help of an expert entomologist from the Academic Biology Unit of the University, where those responsible for the research were assigned.

### 2.5. Mortality Test

In three 250 mL jars (experimental units), 100 g of corn previously exposed to the sun were placed. Subsequently, 1, 2, and 3 g of vegetable powder were added to each jar (1%, 2%, and 3% concentrations, respectively); it was shaken manually using oscillatory and vertical movements for 1 min until a homogeneous powder dispersion was achieved. Once the mixture was made, 20 specimens of adult *S. zeamais* insects were placed in each jar without distinction of sex. Negative control was used without plant powder, and positive control with 0.0018 g of aluminum phosphide. This pesticide was used because it is the chemical product producers use in the community to control the weevil, and it has been used in other studies as a positive control where plant powders have also been evaluated [32].This insecticide is a fumigant tablet purchased in local stores. It comes in a tube with 20 tablets of 3 g each. The concentration of aluminum phosphide is 56%, equivalent to 560 g of A.I./kg, the recommended dose average in corn is 2 to 4 tablets/m^3^. The dose used in the experiment was calculated based on the average volume of corn contained in the jars (196 ± 8.9 cm^3^) from a dose of 3 tablets/m^3^ of aluminum phosphide (in the community, producers use arbitrary doses ranging from 2 to 4 tablets/ton/room), under the same conditions of quantity of corn and insects. This product is applied directly to the grains, and when it comes into contact with ambient humidity, it releases highly toxic phosphine gas, which is responsible for its insecticidal effect [10]. Its effect is rapid, depending on humidity and temperature conditions (24 to 72 h), and its residue is a gray powder that is apparently inert to humans and formed by aluminum hydroxide [33,34]. Still, such residue could harm other organisms, such as earthworms [35]. All jars were covered with organza fabric lids to allow aeration and prevent insect escape. The treatments were placed in a room for 15 days under the typical environmental conditions of temperature, humidity, and community light, separated at 1 m to avoid interaction between them. Exposure was defined for 15 days since the dust’s action was slower than the chemical product. After 15 days, the number of dead adults in each treatment was quantified. The mortality percentage was obtained using the Abbott formula [36]: % Corrected Mortality = [(X − Y)/(100 − Y)] × 100, where X = % Mortality in powder treatment, Y = % Mortality in the negative control (without plant powder). An individual who was not mobile when using a source of heat and light (flashlight) for a minimum of 5 min was considered dead.

### 2.6. Repellency Test

The devices were fabricated using three 250 mL transparent plastic cups connected at their base by a 20 cm long transparent plastic hose (Figure 2). One hundred grams of corn were placed in each side glass (the central glass remained empty). Then, the plant powder was added to one of the glasses with corn, and it was shaken manually using oscillating and vertical movements for 1 min until a homogeneous dispersion of the powder was achieved. The other adjacent cup with corn served as a non-dust control treatment. One device was used per plant and concentration (1%, 2% and 3%). Once the devices were prepared, 30 food-deprived adult insects were released into the central vessel without distinction of sex, aged between 7 and 14 days [37]. Immediately, all the glasses were covered with organza fabric to allow for ventilation and prevent insects from escaping. The devices were placed in the same room and separated at a distance of 1 m to avoid interaction between them. At 24 h after the release of the insects, the number of individuals in each glass was counted; with the values obtained, the repellency index (RI) was calculated using the formula RI = (2G)/(G + P), where G = number of insects in the treatment and P = number of insects in the control. The RI values range from zero to two to obtain the following classification: RI = 1 neutral effect, RI > 1 attracting effect, RI < 1 repellent effect [38].

### 2.7. Experimental Design and Statistical Analysis

The study consisted of two experiments using completely randomized designs with three repetitions each. In the first, 14 treatments were evaluated, corresponding to the powders of the four plant species applied in doses of 1, 2, and 3 g (1%, 2%, and 3%, respectively), plus a control without powder and another with aluminum phosphide. In the second, 12 treatments were evaluated, corresponding to the same concentrations of the plant powders from experiment one, and individually, they were compared against a control without powder. An ANOVA with Tukey’s post hoc test was used to identify differences between treatments in the first experiment. Student’s t-test was used to compare the repellency effect of each vegetable powder by concentration with the control in the second experiment. In both cases, after checking the normality (Shapiro test) and homoscedasticity of variances (Levene’s test) of the data (α = 0.05% and with a confidence level of 95%), the statistical software used was SPSS v.27 (IBM, Armonk, New York, USA).

## 3. Results

### 3.1. Insecticidal Effect

The 3% concentrations of the treatments with *C. ambrosioides* and *P. auritum* were statistically equal after 15 days of exposure, along with the control (aluminum phosphide 0.0018 g), presenting the highest average mortality rates (85.0%, 83.3%, and 100%, respectively) (Table 1). *O. vulgare* and *L. nobilis* at the same concentration of 3%, and *C. ambrosioides* and *P. auritum* at 2%, presented mortalities between 55.0% and 63.3%. The treatments of *O. vulgare* and *L. nobilis* at 2% and the four powders at 1% fluctuated between 21.7% and 36.7% mortality.

### 3.2. Repellency Effect

The treatments with 3% plant powders were statistically equal to each other and different from their corresponding control, showing the highest percentages of repellency between 61.0% and 89.7%. In general, powders at low concentrations (1% and 2%) presented an attractive to neutral effect (RI = 1.7 to 1.0), while for higher concentrations (3%), they presented a repellent effect (RI = 0.8 to 0.2). *L. nobilis* and *O. vulgare* powders stand out for their attractive effect at low concentrations, while *C. ambrosioides* and *P. auritum* powders stand out for their repellent effect at a 3% concentration (Table 2).

## 4. Discussion

In the present study, the best mortality and repellency test results against *S. zeamais* were obtained at a concentration of 3% of *C. ambrosioides* and *P. auritum* powders. The powders of *O. vulgare* and *L. nobilis* at the same concentration presented lower mortality and repellency, and they were not of interest to be included in the present discussion. The findings coincide with various investigations. Regarding the insecticidal and repellent effects of *C. ambrosioides*, in a laboratory study in Paraguay, it was reported that concentrations of 1% to 3% of the powders caused high mortality rates of the insect (81% to 100%) from the 24 h of exposure [39]. In Ethiopia, mortality rates of 100% were obtained at a concentration of 2% to 3% of the powders from the sixth day of exposure [40]. In Cameroon, Central Africa, it was determined that a concentration of 4% caused a mortality rate of 100% from 72 h of exposure to *S. zeamais* [41]. In Yucatán, Mexico, with a concentration of 1%, an RI of 0.51 was obtained after 48 h of exposure against *Zabrotes subfasciatus* Boheman, a weevil that infects *Phaseolus lunatus* L. [42]. Regarding *P. auritum*, there are few studies on its powders’ insecticidal and repellent effects; most studies refer to the effects of its extracts (essential oils) on insects different than *S. zeamais*. In Cuba [43], it was determined that concentrations of 2% powder after 7 days of exposure induced 100% mortality in *Sitophilus oryzae*, a weevil that damages grains of *Pisum sativum* L. It was identified in Choco, Colombia, that after 4 h of exposure, concentrations of 0.2 μL/cm^2^ of essential oil repelled up to 87% of the red flour beetle (*Tribolium castaneum* Herbst) [44]. In Oaxaca, Mexico, it was determined that the concentration of 0.1 g/mL of ethanolic extract induced a mortality of 60% of adult psyllids (*Bactericera cockerelli* [Sulc]; an insect that damages solanaceous plants) after 24 h of exposure [45].

The insecticidal and repellent effects observed in the present study can be attributed to various secondary metabolites in the two main plant species studied. Although its metabolites were not determined, previous studies have already documented them. Approximately 330 compounds, including various isomers, have been identified for *C. ambrosioides* in the different parts of the plant through its essential oils, with the group of monoterpenes predominating, followed by glycosides, flavonoids, esters, aliphatic acids, ketones, aromatic hydrocarbons, carbohydrates, among others [20]. Recently, high concentrations of monoterpene hydrocarbons (63.3% to 78.0%) were determined in the inflorescences and leaves, the main compounds being δ-3-carene, p-cymene, and 1,2:3,4-diepoxy-p-menthane, followed by limonene, β-phellandrene, and γ-terpinene; while oxygenated monoterpenes represented between 19.5% to 31.7%, highlighting the compounds of 1,4-epoxy-p-menth-2-ene, thymol, carvacrol, and ascaridole [46]. The latter has been reported as responsible for its fumigant and contact toxic activity in the control of *S. zeamais* [47]. Regarding *P. auritum*, moderate contents of terpene, coumarin, and flavonoid groups have been identified in ethanolic and aqueous extracts, and low contents of tannins and cardiotonic glycosides [48]. Approximately 27 to 32 metabolites have been reported, mainly predominating safrole, followed by terpinolene, γ-terpinene, β-terpinene, and pinene [49]. Safrole is a toxic component of *P. auritum* [50], to which its insecticidal and repellent effects could be attributed.

During the last decades, the in vitro insecticidal potential of essential oils from various plant species has been demonstrated against several species of insects that cause damage to a variety of crops [51]. Recently, plant powders have gained ground since they are simple to prepare and apply [52] and are easily removable from stored grain without affecting its appearance. The effect of plant powders is mainly explained by causing the death of the individual due to toxicity derived from direct physical contact, making breathing difficult, and/or causing abrasion of the cuticle, leading to water loss and dehydration and/or ingestion; in food deterrence by making the food unattractive and/or unpleasant, causing the death of the insect due to starvation; by decreasing the oviposition rate due to the reduction in the longevity and physical condition of adults to procreate; as well as the presence of volatile compounds that repel individuals [12]. The above translates into a reduction in infestation and a decrease in damage to stored grains.

Despite the results obtained, it is essential to highlight the study’s limitations. Large-scale experiments were not carried out, which could have involved the administration of plant powders in larger quantities of corn and in the conditions of grain storage in the community (generally, the ears and/or grain are stored in sacks, which are settled in a room of the house that serves as a granary). Therefore, it would be necessary to delve into the medium and long-term application strategies to determine the effectiveness and frequency of the doses used in the present study and evaluate the behavior of the fraction of surviving insects. On the other hand, assessing the mixture of both successful plant species could also be considered to determine the probable synergy that improves their insecticidal and repellent potential. It is important to note that the plant species evaluated are common and easily accessible, in addition to being part of the population’s daily diet of all regions of the state of Oaxaca. Therefore, their use, as proposed here in the study, does not represent a risk to public health. It is essential to highlight that the alternative of using plant powders is with the intention of local use, not for commercial purposes. Producers adopting this practice could quickly grow the plants in larger quantities.

Finally, the results obtained are relevant since the study was conducted under the climatic conditions of the rural community of Santa María Zacatepec, Oaxaca, Mexico. This represents enormous potential to replace the various synthetic insecticides (mainly aluminum phosphide) that are being used to prevent damage to corn in storage in the community, in such a way that the risk to the health of farmers and their families can be reduced due to exposure to highly toxic pesticides, in addition to contributing to a healthier environment. It is recommended that in future studies, where the evaluation of these plants is resumed in other places, the determination of metabolites is carried out since edaphic and climatic conditions can influence their production [53], so the application doses could vary.

## 5. Conclusions

*C. ambrosioides* and *P. auritum* powders are potentially ecologically viable bioinsecticides and repellents for controlling *S. zeamais* in stored corn. It is necessary to continue with other studies to evaluate their effectiveness in larger quantities of grain and determine the duration of the insecticidal and repellent effects under rural storage conditions.

## Figures and Tables

**Figure 1 insects-16-00329-f001:**
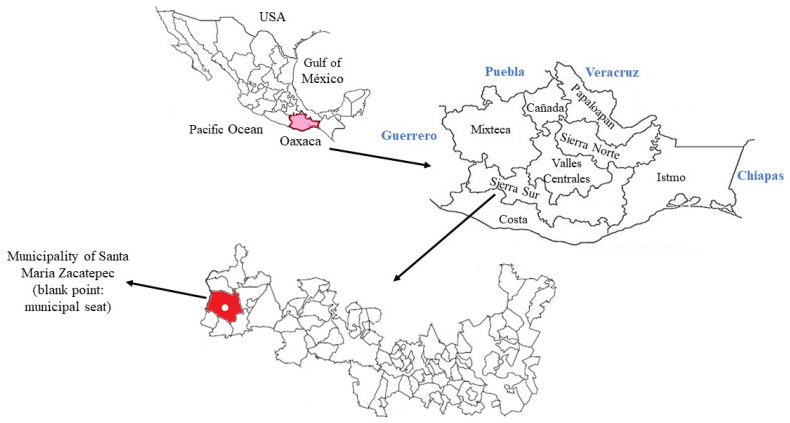
Location of the municipality of Santa María Zacatepec in the Sierra Sur region of Oaxaca, Mexico.

**Figure 2 insects-16-00329-f002:**
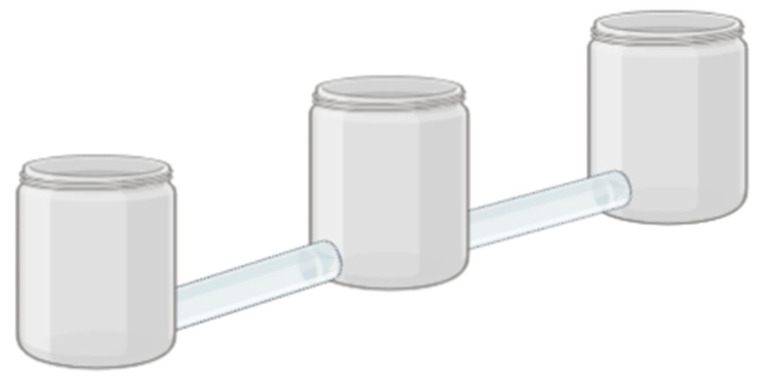
Appearance of the device used in the repellency test.

**Table 1 insects-16-00329-t001:** Mortality of the weevil (*S. zeamais*) due to the effect of different plant powders after 15 days of exposure.

Treatment	Number of Dead Individuals (n = 20) Mean ± SD	Corrected Mortality (%) Mean ± SD
*Origanum vulgare* 1%	4.3 ± 1.2	21.7 ± 5.8 a
*Laurus nobilis* 1%	6.0 ± 2.0	30.0 ± 10.0 ab
*Piper auritum* 1%	6.7 ± 0.6	33.3 ± 2.9 ab
*Chenopodium ambrosioides* 1%	7.3 ± 3.1	36.7 ± 15.3 ab
*Origanum vulgare* 2%	4.7 ± 1.2	23.3 ± 5.8 a
*Laurus nobilis* 2%	6.0 ± 1.0	30.0 ± 5.0 ab
*Piper auritum* 2%	11.0 ± 2.0	55.0 ± 10.0 bc
*Chenopodium ambrosioides* 2%	12.7 ± 1.5	63.3 ± 7.6 cd
*Origanum vulgare* 3%	11.0 ± 2.6	55.0 ± 13.2 bc
*Laurus nobilis* 3%	11.0 ± 1.0	55.0 ± 5.0 bc
*Piper auritum* 3%	16.7 ± 1.5	83.3 ± 7.6 de
*Chenopodium ambrosioides* 3%	17.0 ± 1.7	85.0 ± 8.7 de
Aluminum phosphide	20 ± 0.0	100 ± 0.0 e
Control without vegetable powder	0.0	---

Equal letters in the same column are statistically equal (F = 27.483, *p* = 0.000). The control treatment results without plant powder were not included in the statistical analysis as they caused mortality equal to zero.

**Table 2 insects-16-00329-t002:** Repellency effect of plant powders on *S. zeamais* after 24 h of exposure.

Treatment	No. of Insects Attracted (%) * Mean ± SD	No. of Insects in the Control (%) Mean ± SD	Comparison Treatment Versus Control	Repellency Index (RI) *** Media ± DE	Repellency Effect ***
*Chenopodium ambrosioides* 1% **	18.3 ± 3.5 (61.0 ± 11.5) bcdA	11.7 ± 3.5 (39.0 ± 11.5) A	t = 2.325, *p* = 0.081 ^NS^	1.2 ± 0.3	Attractive
*Piper auritum* 1% **	23.3 ± 3.1 (78.0 ± 10.1) cdA	6.7 ± 3.1 (22.0 ± 10.1) B	t = 6.682, *p* = 0.003 ****	1.5 ± 0.2	Attractive
*Laurus nobilis* 1% **	25.5 ± 3.0 (83.0 ± 10.0) dA	5.0 ± 3.0 (17.0 ± 10.0) B	t = 8.165, *p* = 0.001 ****	1.7 ± 0.2	Attractive
*Origanum vulgare* 1% **	24.3 ± 1.2 (81.0 ± 3.5) dA	5.7 ± 1.2 (19.0 ± 3.5) B	t = 19.799, *p* = 0.001 ****	1.6 ± 0.1	Attractive
*Chenopodium ambrosioides* 2% **	15.3 ± 6.5 (51.0 ± 21.5) bcA	14.7 ± 6.5 (49.0 ± 21.5) A	t = 0.125, *p* = 0.906 ^NS^	1.0 ± 0.5	Neutral
*Piper auritum* 2% **	14.7 ± 3.5 (49.0 ± 11.5) bcA	15.3 ± 3.5 (51.0 ± 11.5) A	t = −0.232, *p* = 0.828 ^NS^	1.0 ± 0.3	Neutral
*Laurus nobilis* 2% **	18.0 ± 2.0 (60.0 ± 7.0) bcdA	12.0 ± 2.0 (40.0 ± 7.0) B	t = 3.674, *p* = 0.021 ****	1.2 ± 0.1	Attractive
*Origanum vulgare* 2% **	16.0 ± 2.0 (53.3 ± 6.5) bcdA	14.0 ± 2.0 (46.7 ± 6.5) A	t = 1.225, *p* = 0.288 ^NS^	1.1 ± 0.1	Attractive
*Chenopodium ambrosioides* 3% **	3.0 ± 1.7 (10.3 ± 5.8) aA	27.0 ± 1.7 (89.7 ± 5.8) B	t= −16.971, *p* = 0.001 ****	0.2 ± 0.1	Repellent
*Piper auritum* 3% **	5.3 ± 2.5 (18.0 ± 8.5) aA	24.7 ± 2.5 (82.0 ± 8.5) B	t= −9.409, *p* = 0.001 ****	0.3 ± 0.2	Repellent
*Laurus nobilis* 3% **	11.7 ± 2.1 (39.0 ± 7.2) abA	18.3 ± 2.1 (61.0 ± 7.2) B	t= −3.922, *p* = 0.017 ****	0.8 ± 0.1	Repellent
*Origanum vulgare* 3% **	11.0 ± 2.0 (36.7 ± 6.5) abA	19.0 ± 2.0 (63.3 ± 6.5) B	t= −4.899, *p* = 0.008 ****	0.7 ± 0.2	Repellent

* Equal lowercase letters in the column “No. of insects attracted” are statistically equal (F = 15.701, *p* = 0.001). ** Equal capital letters in the same row are statistically equal. *** Repellency Index (RI) = RI = 1 neutral effect, RI > 1 attractive effect, RI < 1 repellent effect. **** Student’s *t*-test = *p* < 0.05; NS = not significant.

## Data Availability

All data generated or analyzed during this study are included in this publication.

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
