# Peer review of "Insecticidal and Repellent Activity of Plant Powders on the Weevil (Sitophilus zeamais) in Stored Corn Grains in a Rural Community of Oaxaca, Mexico"

_insects, 2025, doi:10.3390/insects16030329_

Round 1
Reviewer 1 Report
Comments and Suggestions for Authors
Sitophilus zeamais is one of the leading pest insects in grains storage, the bioinsecticidal and repellent effect of four plants powders to S. zeamais was measured and found 3% Chenopodium ambrosioides and Piper auritum have highest mortality and repellent effect. It is an interesting study and their results could be used as a practical guide in stored corn. However, this manuscript could be received to publish in this journal, as it should provide more knowledge to reader or gave further research in mechanism of bioinsecticidal and repellent effect.
Author Response
The reviewers' comments and observations were considered to improve the manuscript.
Reviewer 2 Report
Comments and Suggestions for Authors
The manuscript described about the study of insecticidal and repellent effects of Chenopodium ambrosioides, Origanum vulgare, Piper auritum, and Laurus nobilis based powders against Sitophilus zeamais with corn seeds. Study background information was clearly presented in the introduction section. However, the study is preliminary and lacking of scientific merit. Authors may revise the manuscript with respect to the following comments;
Is it the study (experiments) was performed at rural region or at the laboratory?. Please revise the manuscript and describe clearly.
In the manuscript, lacking of edible, medicinal and safety information’s of the selected plant species in regard to the admixture with edible corn.
Why aluminum phosphide (i.e., phosphine gas) was used as positive control for comparative analysis with the efficacy of plant powders?. Principally, phosphine gas has been used as fumigant not contact toxicant. Is it plant powders cause fumigation effect?. Please justify with relevant literature for the fumigant action of plant powders and application of aluminum phosphide as positive control to plant powders.
Line 2-4: Is it this manuscript study is based on field experiments or laboratory experiments?. With respect to the presented methodology section and laboratory experiments, please remove ‘in a Rural Community of Oaxaca, Mexico’ from the title. Or, revise the methodology section in detail.
Line 14-22: Why summary and abstracts are required?. Please check the journal guidelines. If summary not mandatory, authors may remove the summary section.
Line 23-25: Is it the experimental bioassays were performed at rural houses of study regions or at laboratory?. If performed at rural houses of study region, explain in detail in the methodology section. Or, remove ‘in the rural community of Santa María Zacatepec, Oaxaca, Mexico’.
Line 32: The presented experimental results are not based on the study region. Plant species only collected from the specified area. Accordingly, please remove ‘in rural communities of Oaxaca’.
Line 61-62: With reference to post-harvest storage and indoor applications, how the chemical pesticides accumulate in water and soil?. Please check with the cited reference and justify/revise.
Line 70: What means ‘last presentation’?
Line 74: Is it oviposition or oviposition deterrence?
Line 74-76: How the Chenopodium ambrosioides, Origanum vulgare, Piper auritum, and Laurus nobilis species are more effective than other plant species?. Justify with relevant literature.
Line 83: With respect to the presented methodology, please remove ‘during post-harvest handling in said rural town’.
Line 85-90: Is it the experimental bioassays were performed at rural houses of community of Santa María Zacatepec, located in the Sierra Sur region of Oaxaca, Mexico or at the institute ‘Benito Juárez Autonomous University of Oaxaca’?. Study location mapping not required for the laboratory experimental bioassays. Please remove the section 2.1 and Figure 1. Or, justify with relevant literature.
Line 102-103: If the specimens already deposited, add herbarium reference number(s). If not, add the details of authentication of species identification (taxonomist detail or literature reference).
Line 119-122: Where the insect cultures were maintained?. Please specify clearly in this section.
Line 122-123: Please specify the age of insects, which used for experiments.
Line 129: How the selected plant species powder was considered as vegetable powder?. Add details (vegetable or medicinal properties) in introduction section.
Line 134: Add procurement details of aluminum phosphide.
Line 134: How the 0.0018 g of aluminum phosphide sample was prepared from the tablet?. Explain with relevant reference.
Line 133-137: Is it direct expose/apply of aluminum phosphide is allowed to food grains (i.e., mixing of aluminum phosphide powder directly to corn)?. Please justify with reference.
Line 137-138: If aeration allowed, how aluminum phosphide acted as fumigant?. If not, how aluminum phosphide acted as contact toxicant inside the jars?. Please justify.
Line 133-138: After the 15 days of treatment, how the aluminum phosphide residues were removed from the treated corns?. Add details with relevant references. If not, explain about how the aluminum phosphide treated corns are safer?. Add details with relevant references.
Line 138: Why insect mortality was observed after 15 days?. Is it 15 days is applicable to the phosphine positive control also?. Please justify with relevant references.
Line 149-154: Repellency treatment not clear. Out of 3 jars, in which jar treatment powder was added and which jar was control?. Please specify clearly.
Line 213-214: Please remove the sentence ‘Therefore, they were ………………………. included in the present discussion.’. Not required.
Line 276-277: Please add data of climatic conditions of the rural community of Santa María Zacatepec, Oaxaca, Mexico. In results section, explain the present study experimental conditions comparatively with climatic conditions of the rural community. Then, discuss the both results comparatively in discussion section. Or, remove the sentence ‘Finally, the results obtained are …………………….. of Santa María Zacatepec, Oaxaca, Mexico.’.
Line 285-286: Remove ‘in rural environments in Oaxaca, Mexico’.
Check the binomial names of plant and insect species throughout the manuscript. First time only write full name (e.g., Sitophilus zeamais and Chenopodium ambrosioides) and second time onwards write genus with single letter and species name (e.g., S. zeamais and C. ambrosioides).
Author Response
Is it the study (experiments) was performed at rural region or at the laboratory?. Please revise the manuscript and describe clearly.
Answer: The study was conducted in a rural community, as highlighted in the introduction and methodology.
In the manuscript, lacking of edible, medicinal and safety information’s of the selected plant species in regard to the admixture with edible corn.
Answer: The requested information was included.
Why aluminum phosphide (i.e., phosphine gas) was used as positive control for comparative analysis with the efficacy of plant powders?. Principally, phosphine gas has been used as fumigant not contact toxicant. Is it plant powders cause fumigation effect?. Please justify with relevant literature for the fumigant action of plant powders and application of aluminum phosphide as positive control to plant powders.
Answer: The use of aluminum phosphide was justified in the methodology section.
Line 2-4: Is it this manuscript study is based on field experiments or laboratory experiments?. With respect to the presented methodology section and laboratory experiments, please remove ‘in a Rural Community of Oaxaca, Mexico’ from the title. Or, revise the methodology section in detail.
Answer: The study was carried out in a rural community. Therefore, it is necessary to mention it in the title, and it is detailed in the methodology.
Line 14-22: Why summary and abstracts are required?. Please check the journal guidelines. If summary not mandatory, authors may remove the summary section.
Answer: The Simple Summary and Abstract sections are part of the journal's editorial standards, so they cannot be eliminated,
Line 23-25: Is it the experimental bioassays were performed at rural houses of study regions or at laboratory?. If performed at rural houses of study region, explain in detail in the methodology section. Or, remove ‘in the rural community of Santa María Zacatepec, Oaxaca, Mexico’.
Answer: It was clarified in the Methodology section (2.1) that the experiment was carried out in a house in a rural town.
Line 32: The presented experimental results are not based on the study region. Plant species only collected from the specified area. Accordingly, please remove ‘in rural communities of Oaxaca’.
Answer: The phrase "in the rural communities of Oaxaca" was replaced by "in the rural community where the study was carried out."
Line 61-62: With reference to post-harvest storage and indoor applications, how the chemical pesticides accumulate in water and soil?. Please check with the cited reference and justify/revise.
Answer: This paragraph was kept as is since it refers to pesticides in a general way.
Line 70: What means ‘last presentation’?
Answer: The phrase “Last presentation” was replaced by “Powder presentation”.
Line 74: Is it oviposition or oviposition deterrence?
Answer: It was clarified with the phrase “oviposition deterrent properties”.
Line 74-76: How the Chenopodium ambrosioides, Origanum vulgare, Piper auritum, and Laurus nobilis species are more effective than other plant species?. Justify with relevant literature.
Answer: The last paragraph of the Introduction section mentions that plants are abundant in the region, and their bioinsecticidal properties have been reported in various studies, so they were selected and included in the present study. We consider that the requested justification is found.
Line 83: With respect to the presented methodology, please remove ‘during post-harvest handling in said rural town’.
Answer: The requested phrase was deleted.
Line 85-90: Is it the experimental bioassays were performed at rural houses of community of Santa María Zacatepec, located in the Sierra Sur region of Oaxaca, Mexico or at the institute ‘Benito Juárez Autonomous University of Oaxaca’?. Study location mapping not required for the laboratory experimental bioassays. Please remove the section 2.1 and Figure 1. Or, justify with relevant literature.
Answer: It was clarified that the experiment was carried out in a house in a rural town.
Line 102-103: If the specimens already deposited, add herbarium reference number(s). If not, add the details of authentication of species identification (taxonomist detail or literature reference).
Answer: The literature reference where the taxonomic information is found was added.
Line 119-122: Where the insect cultures were maintained?. Please specify clearly in this section.
Answer: An explanation of how the insects were kept was included.
Line 122-123: Please specify the age of insects, which used for experiments.
Answer: The age of the insects was indicated.
Line 129: How the selected plant species powder was considered as vegetable powder?. Add details (vegetable or medicinal properties) in introduction section.
Answer: Added the information requested in the introduction section.
Line 134: Add procurement details of aluminum phosphide.
Answer: Added the requested information.
Line 134: How the 0.0018 g of aluminum phosphide sample was prepared from the tablet?. Explain with relevant reference.
Answer: Added the requested information.
Line 133-137: Is it direct expose/apply of aluminum phosphide is allowed to food grains (i.e., mixing of aluminum phosphide powder directly to corn)?. Please justify with reference.
Answer: Added the requested information.
Line 137-138: If aeration allowed, how aluminum phosphide acted as fumigant?. If not, how aluminum phosphide acted as contact toxicant inside the jars?. Please justify.
Answer: Added the requested information.
Line 133-138: After the 15 days of treatment, how the aluminum phosphide residues were removed from the treated corns?. Add details with relevant references. If not, explain about how the aluminum phosphide treated corns are safer?. Add details with relevant references.
Answer: Added the requested information.
Line 138: Why insect mortality was observed after 15 days?. Is it 15 days is applicable to the phosphine positive control also?. Please justify with relevant references.
Answer: Added the requested information.
Line 149-154: Repellency treatment not clear. Out of 3 jars, in which jar treatment powder was added and which jar was control?. Please specify clearly.
Answer: The wording was improved to make the procedure more understandable.
Line 213-214: Please remove the sentence ‘Therefore, they were ………………………. included in the present discussion.’. Not required.
Answer: The requested suggestion was applied.
Line 276-277: Please add data of climatic conditions of the rural community of Santa María Zacatepec, Oaxaca, Mexico. In results section, explain the present study experimental conditions comparatively with climatic conditions of the rural community. Then, discuss the both results comparatively in discussion section. Or, remove the sentence ‘Finally, the results obtained are …………………….. of Santa María Zacatepec, Oaxaca, Mexico.’.
Answer: The climatic conditions of the community are mentioned in the Methodology.
Line 285-286: Remove ‘in rural environments in Oaxaca, Mexico’.
Answer: The requested phrase was removed.
Changes were made to the binomial names of plant and insect species throughout the manuscript, the first time using the full name (e.g., Sitophilus zeamais and Chenopodium ambrosioides) and the second time and successively, the single-letter genus and species name (e.g., S. zeamais and C. ambrosioides).
Reviewer 3 Report
Comments and Suggestions for Authors
The manuscript "Insecticidal and Repellent Activity of Plant Powders on the Weevil (Sitophilus zeamais) in Stored Corn Grains in a Rural Community of Oaxaca, Mexico" aimed at evaluating the insecticidal and repellent effect of four plant powders (Chenopodium ambrosioides, Origanum vulgare, Piper auritum, and Laurus nobilis) in the control of adults of Sitophilus zeamais in stored corn during post-harvest handling in said rural town. In this work, researchers investigated the simplest way to protect stored grains, obtained by drying and grinding the plant to mix it with the grain. Since the plant species used in powders have insecticidal properties, and these species are very popular in Mexican cuisine, researchers suggest the technology as a solution to control the weevil in corn storage.
General comments:
- Plant origin insecticides, and repellents have attracted many researchers due their two common features: a- their availability locally in the grain growing ares; and b- due their low environmental impact. However, many of those plant insecticides have at the same time several draw backs: a- it is very difficult to commercialize the product; b- it has a very low commercial interest; c- cost of production is generally very high; d- in many countries they require registration as a pesticides. Those aspects were not considered in the discussion section of the study in the context of the potential application in Mexico.
- A common problem related to the effectiveness of the plant insecticides is in identifying the effective chemical fraction/s of the plant selected for the control of insects. This aspect of using plant insecticides is one of the intriguing topics because even the same plant species under a different climate regime may not produce the chemical fraction that is sufficiently effective to control the insect. This point also should be discussed to assist the reader in being more careful in the application of the technology.
- The suggested solution even under best production conditions is limited to the plant insecticides grown areas. Its production cost would not permit its use at a commercial scale in any other part of the world. It is based on a concentration of powdered leaves of 3% of the stored corn. That means compared to phosphine, which the recommended dosage is about 3 g/tonne, using the powdered botanical will require 30,000 g/tonne. Those are points that a grower should consider before using the botanical solution. The discussion section should refer to those critical points before recommending the botanical solution.
- A very important point is also the limited control achieved using the plant insecticides. The maximum kill was 85% of the population. The remaining 15% live population may still cause a reproduction problem by the time the effect of the botanical is diminished. Researchers did not indicate the necessity of long-term studies to predict the fate of the surviving population when there is no effect of the plant insecticides.
Specific comments:
Line 17: Sugget changing "…is gaining importance…" to "…is gaining attention…"
Line 17: Suggest changing ",, ecologically safer…" to ",, ecologically safer than chemical solutions…"
Line 57: Suggest changing "…oral exposure …" to "…accidental excessive oral exposure…"
Line 108: "…was exposed to the sun for 8 hours…" "… This procedure guaranteed that the corn was free of eggs and adult …" The reason of exposing to the sun should explained. This procedure apparently increased the temperature above 50oC for several hours, sufficient to control all insects life. If temperature of grain was measured it is a better explanation.
Line 140: The time must be quantified. How long time after the 15 days treatment insect mortality was examined? The term "…After time,…" is not sufficient.
Lines 140-141: Researchers must have counted obly adults. If this is the case sentence should be changed from "…number of dead individuals…" to "…number of dead adults…"
Line 149: change "…by a 20 cm transparent…" to "…by a 20 cm long transparent…"
Line 149: change "…hundred grams of…" to "…hundred g of…"
Line 183: change here and in all subsequent mentions of "… Chenopodium ambrosioides…" to "… C. ambrosioides…"
Lines 183-184: change here and in all subsequent mentions of "… Piper auritum…" to "… P. auritum …"
Line 186: change here and in all subsequent mentions of "… Origanum vulgare …" to "… O. vulgare …"
Line 186: change here and in all subsequent mentions of "… Laurus nobilis …" to "… L. nobilis …"
Line 191: if researchers have examined larvae and pupae mortality do not change. But if only adults were examined, change "… of the weevil…" to "… of the adults…"
Line 285: change "…Sitophilus zeamais…" to "…S. zeamais…"
Author Response
Plant origin insecticides, and repellents have attracted many researchers due their two common features: a- their availability locally in the grain growing ares; and b- due their low environmental impact. However, many of those plant insecticides have at the same time several draw backs: a- it is very difficult to commercialize the product; b- it has a very low commercial interest; c- cost of production is generally very high; d- in many countries they require registration as a pesticides. Those aspects were not considered in the discussion section of the study in the context of the potential application in Mexico.
Answer: The discussion included that vegetable powders are proposed for local use, not commercial purposes; therefore, their commercial registration is not required.
A common problem related to the effectiveness of the plant insecticides is in identifying the effective chemical fraction/s of the plant selected for the control of insects. This aspect of using plant insecticides is one of the intriguing topics because even the same plant species under a different climate regime may not produce the chemical fraction that is sufficiently effective to control the insect. This point also should be discussed to assist the reader in being more careful in the application of the technology.
Answer: Discussion related to the relationship between climate and metabolite production was included.
The suggested solution even under best production conditions is limited to the plant insecticides grown areas. Its production cost would not permit its use at a commercial scale in any other part of the world. It is based on a concentration of powdered leaves of 3% of the stored corn. That means compared to phosphine, which the recommended dosage is about 3 g/tonne, using the powdered botanical will require 30,000 g/tonne. Those are points that a grower should consider before using the botanical solution. The discussion section should refer to those critical points before recommending the botanical solution.
Answer: The discussion on the production capacity of the plant species proposed for use as bioinsecticide was included.
A very important point is also the limited control achieved using the plant insecticides. The maximum kill was 85% of the population. The remaining 15% live population may still cause a reproduction problem by the time the effect of the botanical is diminished. Researchers did not indicate the necessity of long-term studies to predict the fate of the surviving population when there is no effect of the plant insecticides.
Answer: The recommendation for medium- and long-term studies regarding the effectiveness of the bioinsecticidal and repellent effects of the powders was included.
Specific comments:
Line 17: Sugget changing "…is gaining importance…" to "…is gaining attention…"
Answer: The suggested change was made.
Line 17: Suggest changing ",, ecologically safer…" to ",, ecologically safer than chemical solutions…"
Answer: The suggested change was made.
Line 57: Suggest changing "…oral exposure …" to "…accidental excessive oral exposure…"
Answer: The suggested change was made.
Line 108: "…was exposed to the sun for 8 hours…" "… This procedure guaranteed that the corn was free of eggs and adult …" The reason of exposing to the sun should explained. This procedure apparently increased the temperature above 50oC for several hours, sufficient to control all insects life. If temperature of grain was measured it is a better explanation.
Answer: The requested explanation was made.
Line 140: The time must be quantified. How long time after the 15 days treatment insect mortality was examined? The term "…After time,…" is not sufficient.
Answer: The requested explanation was made.
Lines 140-141: Researchers must have counted obly adults. If this is the case sentence should be changed from "…number of dead individuals…" to "…number of dead adults…"
Answer: The suggested change was made.
Line 149: change "…by a 20 cm transparent…" to "…by a 20 cm long transparent…"
Answer: The suggested change was made.
Line 149: change "…hundred grams of…" to "…hundred g of…"
Answer: The suggested change was not made since the numerical value is not used, and changing the word “grams” to its symbol “g” could generate errors in understanding the text. However, if the editor considers it pertinent, we have no objection to correcting.
Line 191: if researchers have examined larvae and pupae mortality do not change. But if only adults were examined, change "… of the weevil…" to "… of the adults…"
Answer: The suggested change was made.
Changes were made to the binomial names of plant and insect species throughout the manuscript, the first time using the full name (e.g., Sitophilus zeamais and Chenopodium ambrosioides) and the second time and successively, the single-letter genus and species name (e.g., S. zeamais and C. ambrosioides).
Round 2
Reviewer 2 Report
Comments and Suggestions for Authors
Authors were revised the manuscript and they responded to the raised comments. However, the revised manuscript and authors responses are not satisfactory. Still, the manuscript contains major errors. Especially, lacking of information on authentication for the studied plants species, positive control experimentation details, detailed methodologies for the experiments performed at houses, and results presentations. The manuscript needs major revision. Major comments follows;
Line 95-97: How the experiments were performed at rural houses?. Lacking of detailed and scientific methodologies (number of houses?, number of samples per house?) with supportive references/earlier studies citations.
Still authors responses are not satisfactory to the raised questions ‘Why aluminum phosphide (i.e., phosphine gas) was used as positive control for comparative analysis with the efficacy of plant powders?. Principally, phosphine gas has been used as fumigant not contact toxicant. Is it plant powders cause fumigation effect?. Please justify with relevant literature for the fumigant action of plant powders and application of aluminum phosphide as positive control to plant powders.
How the plant powders acted as fumigant as similar to the positive control?. Lacking of justification with relevant literature in the revised manuscript.
Line 104-110: If the experiments were performed at local houses (check line 95-97), how the plant materials were weighed (0.5 Kg)?, how the room temperature was measured?, and why the powdered plant materials were conserved?. Lacking of procurement details of sieve (297 μm mesh).
Line 110-112: Plant species identification and authentication details are not clear. Authors should add the taxonomist details and herbarium reference numbers for the authentication of collected plant species.
Line 140: ‘Previously treated corn’ means?
Line 146-147: Aluminium phosphide (ALP) procurement details (manufacturer, item code, quality standard and specific place) are not clear. How it is available in local shop/seller?. Which manufacturer recommended the dosage ‘2 to 4 tablets/m3?.
Line 153: How the applying ALP directly to the food grains is scientifically correct?. Phosphine is an residue free-fumigant, but how residue from ALP tablet is safe?. Please refer ‘Wang et al., (2014)’. If applied directly, what about the residue contamination risks in grains from aluminium phosphide tablet?. Lacking of supportive literature information or similar studies citations in the manuscript.
Line 141 & Line 150: For the insecticidal assay, 1 to 3% of plant powders were mixed with grain. But, the positive control ‘phosphine’ concentration is not clear. In table 1, insecticidal effect of plant powders were compared with the positive control ALP. How the test substances (1 to 3%) were compared positive control (conc. ?) without concentration?.
Why two different terms ‘vegetable powder’ and ‘plant powder’ were used for the same material?.
Author Response
Reviewer 2
Line 95-97: How the experiments were performed at rural houses?. Lacking of detailed and scientific methodologies (number of houses?, number of samples per house?) with supportive references/earlier studies citations.
Answer: The observation was addressed in the Methodology section.
Still authors responses are not satisfactory to the raised questions ‘Why aluminum phosphide (i.e., phosphine gas) was used as positive control for comparative analysis with the efficacy of plant powders?. Principally, phosphine gas has been used as fumigant not contact toxicant. Is it plant powders cause fumigation effect?. Please justify with relevant literature for the fumigant action of plant powders and application of aluminum phosphide as positive control to plant powders.
Answer: The observation was addressed in the Methodology section.
How the plant powders acted as fumigant as similar to the positive control?. Lacking of justification with relevant literature in the revised manuscript.
Answer: The observation was addressed in the Introduction section.
Line 104-110: If the experiments were performed at local houses (check line 95-97), how the plant materials were weighed (0.5 Kg)?, how the room temperature was measured?, and why the powdered plant materials were conserved?. Lacking of procurement details of sieve (297 μm mesh).
Answer: The observation was addressed in the Methodology section.
Line 110-112: Plant species identification and authentication details are not clear. Authors should add the taxonomist details and herbarium reference numbers for the authentication of collected plant species.
Answer: The observation was addressed in the Methodology section.
Line 140: ‘Previously treated corn’ means?
Answer: It was changed to the phrase: corn previously exposed to the sun.
Line 146-147: Aluminium phosphide (ALP) procurement details (manufacturer, item code, quality standard and specific place) are not clear. How it is available in local shop/seller?. Which manufacturer recommended the dosage ‘2 to 4 tablets/m3?.
Answer: The observation was addressed in the Methodology section.
Line 153: How the applying ALP directly to the food grains is scientifically correct?. Phosphine is an residue free-fumigant, but how residue from ALP tablet is safe?. Please refer ‘Wang et al., (2014)’. If applied directly, what about the residue contamination risks in grains from aluminium phosphide tablet?. Lacking of supportive literature information or similar studies citations in the manuscript.
Answer: The observation was addressed in the Methodology section.
Line 141 & Line 150: For the insecticidal assay, 1 to 3% of plant powders were mixed with grain. But, the positive control ‘phosphine’ concentration is not clear. In table 1, insecticidal effect of plant powders were compared with the positive control ALP. How the test substances (1 to 3%) were compared positive control (conc. ?) without concentration?.
Answer: Since there was no mortality in the dust-free control (the values ​​were zero), comparing it with the rest of the treatments was unnecessary. This note is made at the bottom of Table 1.
Why two different terms ‘vegetable powder’ and ‘plant powder’ were used for the same material?.
Answer: The manuscript was homogenized with the term “plant powders”
Round 3
Reviewer 2 Report
Comments and Suggestions for Authors
The revised manuscript is improved. However, please consider the following minor corrections in the manuscript;
Please remove figure 2 “Appearance of the tube with aluminium phosphide tablets sold in the community of Santa María Zacatepec, Oaxaca, Mexico.”
Line 159-160: Please remove the sentences “Its label only contains information about the product's name, not its use (Fig. 2). Therefore, several technical sheets of products authorized and marketed for Mexico are consulted to identify their characteristics [33-35]”.
Author Response
Please remove figure 2 “Appearance of the tube with aluminium phosphide tablets sold in the community of Santa María Zacatepec, Oaxaca, Mexico.”
Answer: Figure 2 and its corresponding citation in the text were removed.
Line 159-160: Please remove the sentences “Its label only contains information about the product's name, not its use (Fig. 2). Therefore, several technical sheets of products authorized and marketed for Mexico are consulted to identify their characteristics [33-35]”.
Answer: The sentence and its corresponding references were removed.
The numbering of the references was corrected.